# MiR-146a Is Mutually Regulated by High Glucose-Induced Oxidative Stress in Human Periodontal Ligament Cells

**DOI:** 10.3390/ijms251910702

**Published:** 2024-10-04

**Authors:** Chihiro Fumimoto, Nobuhiro Yamauchi, Emika Minagawa, Makoto Umeda

**Affiliations:** Department of Periodontology, Osaka Dental University, 8-1, Kuzuhahanazono-cho, Hirakata 573-1121, Osaka, Japan; fumimoto-c@cc.osaka-dent.ac.jp (C.F.); minagawa-e@cc.osaka-dent.ac.jp (E.M.); umeda-m@cc.osaka-dent.ac.jp (M.U.)

**Keywords:** miR-146a, hPDLCs, ROS, high glucose

## Abstract

The high-glucose conditions caused by diabetes mellitus (DM) exert several effects on cells, including inflammation. miR-146a, a kind of miRNA, is involved in inflammation and may be regulated mutually with reactive oxygen species (ROS), which are produced under high-glucose conditions. In the present study, we used human periodontal ligament cells (hPDLCs) to determine the effects of the high-glucose conditions of miR-146a and their involvement in the regulation of oxidative stress and inflammatory cytokines using Western blotting, PCR, ELISA and other methods. When hPDLCs were subjected to high glucose (24 mM), cell proliferation was not affected; inflammatory cytokine expression, ROS induction, interleukin-1 receptor-associated kinase 1 (IRAK1) and TNF receptor-associated factor 6 (TRAF6) expression increased, but miR-146a expression decreased. Inhibition of ROS induction with the antioxidant N-acetyl-L-cysteine restored miR-146a expression and decreased inflammatory cytokine expression compared to those under high-glucose conditions. In addition, overexpression of miR-146a significantly suppressed the expression of the inflammatory cytokines IRAK1 and TRAF6, regardless of the glucose condition. Our findings suggest that oxidative stress and miR-146a expression are mutually regulated in hPDLCs under high-glucose conditions.

## 1. Introduction

Diabetes mellitus (DM) is a major risk factor for periodontal disease [1]. It has been reported that diabetic patients are 2.6 times more likely to develop periodontal disease than non-diabetic patients [2]. In diabetic periodontitis, high glucose due to diabetes has several effects on periodontal tissues. It has been reported that high glucose inhibits the wound healing of periodontal tissue fibroblasts [3]. High glucose induces apoptosis of vascular endothelial cells and inhibits bone differentiation of bone marrow mesenchymal stem cells [4,5]. Human periodontal ligament cells (hPDLCs) are essential cells with different functions in periodontal tissues, including periodontal disease progression, wound healing and tissue regeneration [6]. A previous study reported that high-glucose conditions may induce inflammatory cytokine production in hPDLCs [7].

MicroRNAs (miRNAs) are small noncoding RNAs, only 20 to 25 nucleotides in length and are involved in post-transcriptional regulation [8]. These small RNAs target multiple messenger RNAs (mRNAs) and mediate mRNA degradation by binding to the 3` untranslated region (UTR) [8,9,10]. The regulation of miRNAs, which negatively control target gene expression, is involved in many physiological processes, such as cell growth and differentiation [11]. MiRNA (miR)-146a is a member of the miR-146 family and has been reported in the past to be involved in the onset and progression of various diseases, including osteoporosis and Alzheimer’s disease [12,13]. Extensive studies have indicated targeting interleukin-1 receptor-associated kinase 1 (IRAK1) and TNF receptor-associated factor 6 (TRAF6) as regulatory mechanisms of miR-146a may act as an anti-inflammatory strategy [14].

Oxidative stress refers to a state in which the balance of the cell’s redox mechanism is disrupted due to excessive production of reactive oxygen species (ROS) [15]. Overproduction of ROS is recognized as a cause of several disease processes and is a major factor in the inflammatory response to high-glucose conditions caused by DM [16]. Several studies have reported the association between oxidative stress and altered miRNA expression in vitro [17]. It has been reported that ROS induction is increased by the decreased expression of miRNA-146a under high-glucose conditions [18], suggesting the significance of miRNA expression in oxidative stress generation or suppression. In contrast, the intracellular redox status is involved in miRNA expression by affecting its biosynthesis mechanism [19] and the relationship between oxidative stress and miRNA expression may be mutually regulated.

The role of miRNAs in various cell types has been well studied and it has been reported miR-146a is differentially expressed in different tissues in DM [1]. In the past, it has been reported that miR-146a in hPDLCs increases in expression due to induction by infection in inflammatory responses caused by periodontal pathogenic bacteria, but in hyperglycemic states, its expression decreases [20]. We hypothesized that the inflammatory response induced by high glucose may be due to the fact that the intracellular oxidative stress state decreases the expression of miR-146a, which has anti-inflammatory effects. However, the behavior of miR-146a in hPDLCs under high-glucose conditions and its relationship with oxidative stress remain unknown. We investigated the effects of high glucose on miR-146a expression in hPDLCs and the involvement of miR-146a in the regulation of oxidative stress and inflammatory cytokines.

## 2. Results

### 2.1. High-Glucose Conditions Increased Inflammatory Cytokine Production without Affecting Cell Proliferation

We discuss the effect of hPDLC proliferation and the production of inflammatory cytokines (interleukin [IL]-6 and IL-8) under high-glucose conditions. No significant difference was noted in cell proliferation between the control (5.5 mM) and high-glucose (24 mM) groups at 24, 48 and 72 h (Figure 1A–C). IL-6 production was significantly increased at 72 h, whereas IL-8 production and *IL-6* and *IL-8* mRNA expression was significantly increased at 48 and 72 h (Figure 1D–K).

### 2.2. ROS Induction Was Enhanced under the High-Glucose Condition

ROS induction under high-glucose conditions was measured. ROS levels were significantly increased under high-glucose conditions compared to those in the control group at 72 h (Figure 2A,B). Morphological changes in mitochondria were observed using membrane potential-independent dyes by confocal laser microscopy. Long tubular structures in interconnected networks were observed in the control. Under high-glucose conditions, mitochondria were fragmented into short rods or spheres (Figure 2C). NO levels were significantly decreased under high-glucose conditions compared to the control at 72 h (Figure 2D–F).

### 2.3. High-Glucose Conditions Decreased miR-146a Expression and Increased TRAF6 and IRAK1 and NF-κB Expression

The effects of high-glucose conditions on miR-146a, TARF6, IRAK1 and NF-κB expression were investigated (Figure 3A–G). The expression of miR-146a in hPDLCs significantly decreased under high-glucose conditions compared to that under low-glucose conditions (Figure 3A). However, high glucose levels enhanced the expression of interleukin-1 receptor-associated kinase 1 (IRAK1) and the TNF receptor-associated factor 6 (TRAF6) and Nuclear factor-kappa B (NF-κB) proteins, which were inversely proportional to the changes in miR-146a (Figure 3C–G).

### 2.4. Inhibition of ROS Induction Restored miR-146a Expression and Reduced Inflammatory Cytokine Production

N-acetyl-L-cysteine (NAC), an ROS inhibitor, was used to evaluate the effect of ROS induction under high-glucose conditions. The optimal NAC concentration was 5 mM because NAC treatment decreased ROS induction under high-glucose conditions and significantly suppressed it at 5 and 10 mM (Figure 4A). NAC treatment significantly increased the expression of miR-146a, which was decreased under high-glucose conditions (Figure 4B).

NAC treatment significantly decreased IL-6 and IL-8 production and gene expression, which were increased under high-glucose conditions (Figure 4C–F).

### 2.5. Overexpression of miR-146a Decreased Inflammatory Cytokine Production, ROS Induction, and TRAF6, IRAK1 and NF-κB Expression

To investigate the involvement of miR-146a in inflammatory cytokine production, ROS induction, and TRAF6 and IRAK1 expression in hPDLCs, we transfected miR-146a mimic for 24 h. The expression and production of IL-6, IL-8 and ROS significantly decreased in the mimic-treated groups at both 5.5 mM and 24 mM concentrations (Figure 5A–F). Similarly, the expression of IRAK1, TRAF6 and NF-κB proteins significantly decreased in mimic-transfected cells at both 5.5 mM and 24 mM concentrations (Figure 6A–F).

## 3. Discussion

In this study, we observed miR-146a expression in hPDLCs under high-glucose conditions and investigated its regulatory signaling pathways and changes in its expression under oxidative stress conditions. DM causes an inflammatory response in periodontal tissue, increasing the risk of periodontitis [21]. High-glucose conditions increase the levels of inflammatory cytokines and exacerbate the inflammatory response. Increased cytokine levels further alter insulin sensitivity, leading to a vicious cycle of DM progression and periodontal tissue damage [22]. Several glucose concentrations have been used in vitro in the past to assess the effects of high-glucose conditions on DM and these effects vary from cell to cell. In a previous report, human gingival fibroblasts and periodontal ligament cells were affected differently in terms of their proliferative potential at the same concentration of high glucose [23]; therefore, it is important to examine the effects on different cell types. Although glucose concentrations of 30 mM or higher have been studied in the past, well above the blood glucose levels observed in DM, the equivalent glucose concentrations for healthy subjects and DM with poor glycemic control are 5.5 mM (99 mg/dL) and 24 mM (432 mg/dL) [24,25]. These two glucose concentrations were used to examine their effects on hPDLCs in DM. The production of inflammatory cytokines IL-6 and IL-8 increased in hPDLCs under high-glucose conditions without affecting their proliferative capacity. Wu et al. reported that both 15 mM and 25 mM high-glucose conditions exerted no effect on cell proliferation in human gingival fibroblasts; however, these increased inflammatory cytokine production such as that of IL-6 and TNF-α at 25 mM [26], which is consistent with our present results.

MiRNAs, which are RNAs that do not encode proteins, play an important role in post-transcriptional regulation; the relationship between several diseases and miRNAs has been studied [8]. For instance, miR-146a has been demonstrated in several studies to be involved in several inflammatory diseases [27]. It has been implicated in regulating innate immunity and inflammation and is an important factor in both diabetes and periodontitis [28]. The expression of miR-146a has been reported to increase in gingiva with periodontal disease compared to healthy gingiva [29]. Motedayen et al. demonstrated higher miR-146a levels and lower expression of inflammatory cytokines, especially IL-6 and TNF-α, in gingival tissue from patients with chronic periodontitis than in those from healthy controls [30]. In contrast, miR-146a is differentially expressed in different tissues of patients with DM [1] and its expression is decreased in human retinal microvascular endothelial cells [31] and peripheral blood mononuclear cells [32]. Similarly, Feng et al. reported low miR-146a expression and increased inflammatory cytokine production in human cardiac microvascular endothelial cells under high-glucose conditions; however, there has been no study of miR-146a expression in periodontal tissue [33]. In the present study, we demonstrated that the miR-146a expression decreased in hPDLCs under high-glucose conditions, and its overexpression decreased the production of IL-6 and IL-8 induced by high-glucose conditions. In hPDLCs, decreased expression of miR-146a under high-glucose conditions exacerbated inflammatory responses. The investigation of miRNAs may serve as a new strategy for disease diagnosis and treatment. MiRNAs regulate gene expression by binding to the 3’-UTR of their target genes [34]. Binding is not perfect and a single miRNA can bind to multiple mRNAs and target several genes to participate in different cellular processes [5]. TRAF6 and IRAK1, which are important adaptor molecules in the inflammatory signaling pathway, have been reported as targets of the post-translational repression of miR-146a [35]. IRAK1 is a key mediator of the IL-1 receptor (IL-1R) and Toll-like receptor (TLR) pathways that mediate the NF-κB signaling pathway, whereas TRAF6 is a key signaling molecule of the TRAF receptor superfamily and IL-1/TLR superfamily [11]. MiR-146a has been predicted to down-regulate target genes such as *TRAF6* and *IRAK1* to suppress excessive inflammatory responses [36]. And even in the absence of inflammatory stimulation, it has been reported that inhibition of miR-146a expression alone increases inflammatory mediators [37]. Xie et al. reported that miR-146 inhibits the lipopolysaccharide (LPS)-stimulated increase in inflammatory cytokines induced by IRAK1 in human gingival fibroblasts [38]. Shali et al. reported that high-glucose conditions in the renal cortex of mice increased the expression of IRAK1 and TRAF6. However, this expression was suppressed in transgenic mice overexpressing endothelial-specific miR-146a [39]. In the present study, the expression of IRAK1 and TRAF6 increased in hPDLCs under high-glucose conditions and their expression was decreased by the overexpression of miR-146a. This indicates that miR-146a targets TRAF6 and IRAK1 to regulate inflammation in hPDLCs under high-glucose conditions.

ROS is a general term for chemical species containing partially reduced oxygen that are thought to induce cytotoxicity and damage lipids, proteins and DNA [40]. Overproduction of ROS causes oxidative stress with an intracellular redox imbalance. Intracellularly, mitochondria form a reticular network of filamentous tubules [41] and previous reports have shown that this network is fragmented under high-glucose conditions, leading to increased levels of ROS [42]. Several studies suggest that ROS overproduction is a direct result of high glucose and that ROS generated by mitochondria due to high glucose are a major initiator of multiple pathological pathways in diabetes [43,44]. Oxidative stress conditions have been reported in the past to alter miRNA expression [45]. For example, Simone et al. reported that H_2_O_2_ treatment altered the expression of miR-16, miR-21, miR-22, miR-99a, miR-143 and miR-155 in human fibroblasts [46]. Dicer, which plays an important role in miRNA biogenesis, converts pre-miRNAs to active single-stranded miRNAs [47], and it has been reported that H_2_O_2_, a type of ROS, inhibits the expression of Dicer and decreases miRNA expression [19]. In the present study, miR-146a expression decreased in hPDLCs under high-glucose conditions, and its expression was restored by the inhibition of ROS production by NAC. miR-146a expression is affected by oxidative stress caused by high-glucose conditions. MiRNAs have been demonstrated to be involved in the maintenance of the redox state; they weaken oxidative stress through regulation of the antioxidant system and exacerbate it by affecting the ROS-producing system [48,49]. And even in the absence of inflammation induction, it has been reported that inhibition of miR-146a expression alone increases inflammatory cytokine production of periodontal tissue [38]. Previously, miR-146a was reported to be involved in ROS production by targeting NADPH oxidase, an ROS-producing enzyme [50]. Wan et al. reported that the overexpression of miR-146a in HK-2 human kidney cells reduced high glucose-induced ROS generation [51]. Qu et al. demonstrated that the overexpression of miRNA-146a in rat neurons suppressed the TRAF6/NF-κB pathway and reduced inflammatory cytokine production and oxidative stress [52]. In the present study, ROS induction was increased in hPDLCs under high-glucose conditions and the overexpression of miR-146a reduced ROS generation induced by high glucose. This indicates that miR-146a regulates intracellular oxidative stress induced by high glucose. These results confirm our hypothesis that the oxidative stress-induced reduction in miR-146a expression leads to high glucose-induced inflammation of hPDLCs and further suggest that oxidative stress and miR-146a may be regulated in a bidirectional manner in hPDLCs under high-glucose conditions.

The present study, which demonstrates the involvement of miR-146a in the complex inflammatory response of periodontal tissue induced by high-glucose conditions, has potential applications for improving the clinical diagnosis and prognosis, but it is only supported by in vitro experiments. It is also limited to a single miRNA. In high-glucose conditions due to DM, a variety of miRNAs may be involved in complex pathologies. Future studies should be extended to investigate the involvement of other miRNAs and the changes in miRNAs due to complex interactions between tissues using in vivo experiments.

## 4. Materials and Methods

### 4.1. Cell Culture

All PDLCs were obtained with approval in accordance with the Osaka Dental University Medical Ethics Guidelines, and these experiments were approved by the Medical Ethics Committee of Osaka Dental University (Approval no. 111132). This study was conducted in accordance with the Declaration of Helsinki 1975, as revised in 2013. hPDLCs were isolated and cultured as described previously [24]. Briefly, the cells were cultured in Dulbecco’s modified Eagle’s medium (DMEM; Nacalai Tesque, Kyoto, Japan) supplemented with 10% fetal bovine serum (FBS; Hyclone, Thermo Fisher Scientific, Waltham, MA, USA) and antibiotics (100 U/mL penicillin, 100 μg/mL streptomycin and 25 μg/mL amphotericin B; Nacalai Teaque) under 37 °C and 5% CO_2._ PDLCs were cultured with 5.5 mM and 24 mM glucose.

### 4.2. Cell Proliferation Assay

Proliferation data were conducted using the Cell Count Reagent SF (Nacalai Tesque). The absorbance was analyzed using SoftMax Pro Microplate Data Acquisition and Analysis software (Version7.0, Molecular Devices; Sunnyvale, CA, USA). For live staining, the PDLCs were stained with calcein acetoxymethyl ester (Calcein-AM) (Dojindo, Kumamoto, Japan). Stained cells were imaged using a BZ-II all-in-one fluorescence microscope (Keyence Corporation; Osaka, Japan).

### 4.3. Real-Time Polymerase Chain Reaction (PCR)

Total RNA was isolated using the RNeasy Mini Kit (Qiagen; Venlo, The Netherlands) and reverse transcribed into cDNA using the PrimeScript RT Reagent Kit (TAKARA Bio, Shiga, Japan). PCR was performed to determine the expression of IL-6 and IL-8 (TaqMan^®^ Gene Expression Assay; Applied Biosystems™, Thermo Fisher Scientific, Waltham, MA, USA). To evaluate miRNAs, the miRNeasy Mini Kit (Qiagen) was used to extract the total RNA. Next, cDNA was synthesized using TaqMan microRNA assays (Thermo Fisher Scientific) and a microRNA Reverse Transcription Kit (Thermo Fisher Scientific) with specific RT primers for hsa-miR-146a-5p and U6. U6 spliceosomal RNA was used as the internal control.

### 4.4. ELISA

For the quantification of human IL-6 and IL-8, Human IL-6 ELISA Kit (Thermo Fisher Scientific) and Human IL-8 ELISA Kit (Thermo Fisher Scientific) were used, respectively. The assays were performed according to the manufacturer’s instructions.

### 4.5. ROS and NO Detection

ROS levels were assessed under 5.5 mM and 24 mM glucose conditions using a Total ROS Detection Kit (Dojindo). The stained cells were observed and imaged using a BZ-X800 all-in-one fluorescence microscope (Keyence, Osaka, Japan). Cellular fluorescence was measured using version 7.0 SoftMax Pro Microplate Data Acquisition and Analysis software (Molecular Devices). NO was determined using a Griess reagent kit (Sigma Aldrich, MO, USA). Absorbance was read at 540nm.

### 4.6. Western Blot Analysis

hPDLCs were lysed in the RIPA buffer containing the protease inhibitor and phosphatase inhibitor, and we measured the protein concentrations. Protein samples were separated by SDS-PAGE and transferred. Next, the membranes were blocked and incubated with primary antibodies against TRAF6, IRAK1, NF-κB, pNF-κB and β-actin (Cell Signaling Technology; Danvers, MA, USA). The membranes were washed and incubated with secondary antibodies. The immunoreactive bands were made visible using a chemiluminescence kit (Nakarai Tesque) and the signal and Western blot data were analyzed using ImageJ Ver. 1.53e (Wayne Rasband and contributors; National Institutes of Health, USA).

### 4.7. Immunofluorescence Staining

Following stimulation, cells were incubated with primary antibodies against TRAF6 and IRAK1 (Santa Cruz Biotechnology; Inc. Dallas, TX, USA). Fluorescent immunostaining was performed using Alexa Fluor 488^®^ (Thermo Fisher Scientific) as the secondary antibody and the nuclei were stained using 4’,6-diamidino-2-phenylindole (DAPI) (Dojindo). Images of stained cells were obtained using a fluorescence microscope. The images were analyzed.

### 4.8. Transfection of miRNA Mimics

The mirVana miRNA mimic for hsa-miR-146a-5p (miR-146a mimic; Thermo Fisher Scientific) was transfected into hPDLCs using the Lipofectamine RNAiMAX transfection reagent (Thermo Fisher Scientific). mirVana miRNA mimic Negative Control #1 (miR-146a NC; Thermo Fisher Scientific) was used as the control. The cells were forward-transfected for 24 h according to the manufacturer’s protocol.

### 4.9. Statistical Analysis

Data were analyzed and are presented as mean ± standard deviation (SD). Parametric data were analyzed using one-way analysis of variance (ANOVA) with Tukey’s test using IBM SPSS Statistics Ver. 17 (IBM; Chicago, IL, USA).

## 5. Conclusions

Oxidative stress caused by high-glucose conditions increases the expression of inflammatory cytokines by inhibiting miR-146a expression in hPDLCs. The reduced expression of miR-146a may exacerbate oxidative stress, leading to a vicious cycle and complicating the pathophysiology of DM. This study may help in the early detection and treatment of periodontal disease in DM.

## Figures and Tables

**Figure 1 ijms-25-10702-f001:**
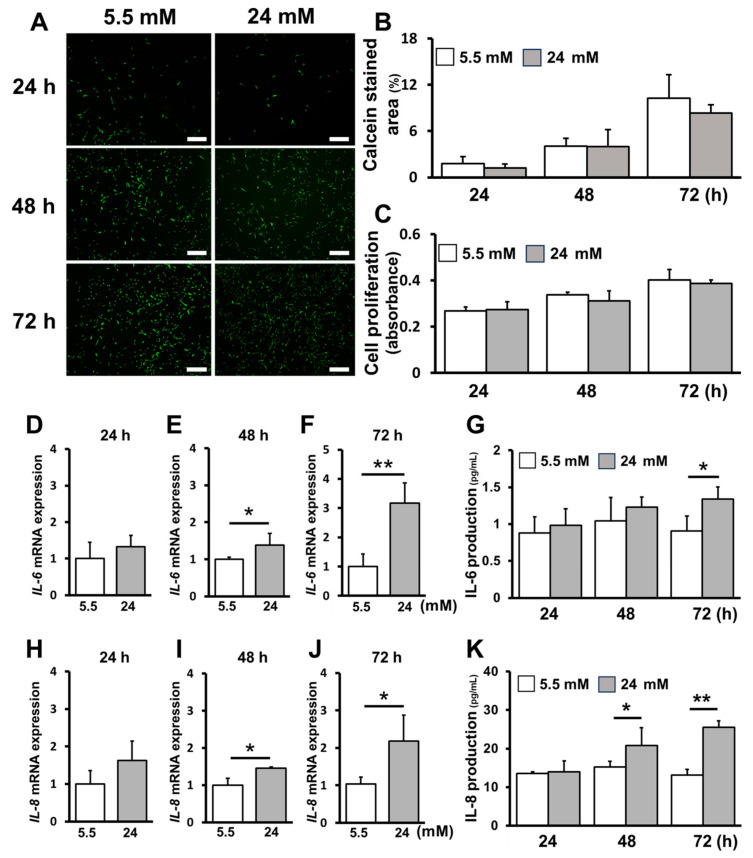
High glucose concentrations increase inflammatory cytokine production without affecting the proliferation of human periodontal ligament cells (hPDLCs). (**A**) hPDLCs stained with calcein-AM were imaged by a fluorescence microscope at 24, 48 and 72 h after incubation (scale bars: 500 μm). (**B**) The data for live cell staining are shown as the percentage of the area stained with calcein. (**C**) Cell proliferation was measured 24, 48 and 72 h after incubation. (**D**–**F**) *IL-6* mRNA gene expression was assessed at the point of 24, 48 and 72 h after stimulation. (**G**) IL-6 production was measured at 24, 48 and 72 h. (**H**–**J**) *IL-8* mRNA gene expression was assessed at the point of 24, 48 and 72 h after stimulation. (**K**) IL-8 production was measured at the point of 24, 48 and 72 h after stimulation. Significant increase control: **p* < 0.05, *** p* < 0.01.

**Figure 2 ijms-25-10702-f002:**
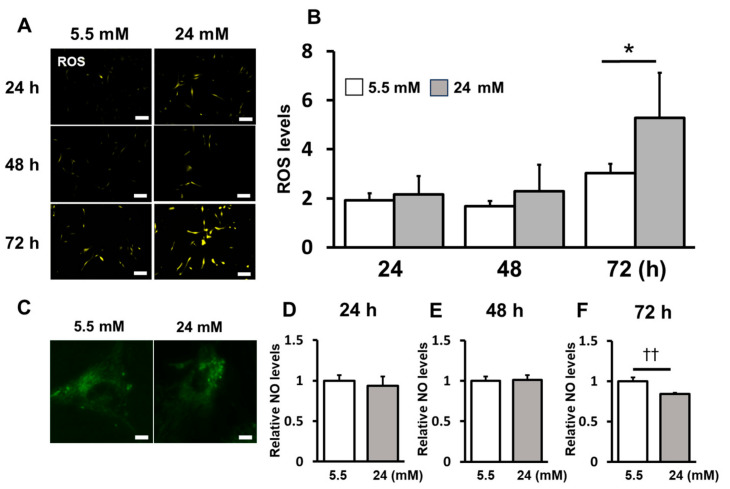
ROS induction was enhanced under high-glucose conditions. (**A**) The fluorescence intensity of ROS levels was measured using a plate reader and the data were compared to those of the control (scale bar: 200 μm). (**B**) Fluorescent staining was performed using a fluorescence microscope. (**C**) The mitochondria were stained after stimulation using MitoTracker (scale bar: 50 μm). (**D**–**F**) NO production was measured using a Griess assay at 24, 48 and 72 h after stimulation. Significant increases compared with the control: ** p* < 0.05, ††*p* < 0.05.

**Figure 3 ijms-25-10702-f003:**
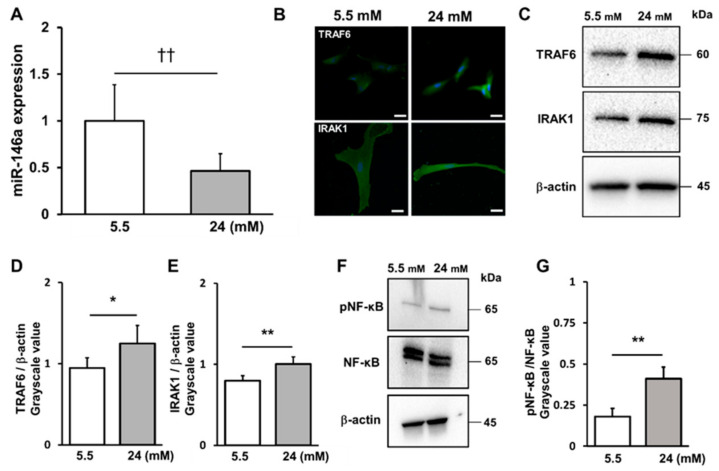
High-glucose conditions decrease miR-146a expression and increase IRAK1, TRAF6 and NF-kB expression. (**A**) MiR-146a expression was measured at 72 h. (**B**) Immunofluorescence staining of IRAK1 and TRAF6 was visualized by confocal laser microscopy after 72 h of incubation (scale bar: 50 μm). (**C**) The levels of IRAK1 and TRAF6 were analyzed using Western blotting. Western blotting was performed on protein extracts of these cells with antibodies against the indicated proteins, using β-actin as a loading control. (**D**) TRAF6 expression was quantified using ImageJ software. (**E**) IRAK1 expression was quantified using ImageJ software (version 1.53e). (**F**) NF-kB and pNF-kB were analyzed using Western blotting. Western blotting was performed on protein extracts of these cells with antibodies against the indicated proteins, using β-actin as a loading control. (**G**) NF-kB and pNF-kB expression was quantified using ImageJ software. Significant increases compared with the control: ** p* < 0.05, *** p* < 0.01. Significant decreases compared with those at 5.5 mM: ††* p* < 0.01.

**Figure 4 ijms-25-10702-f004:**
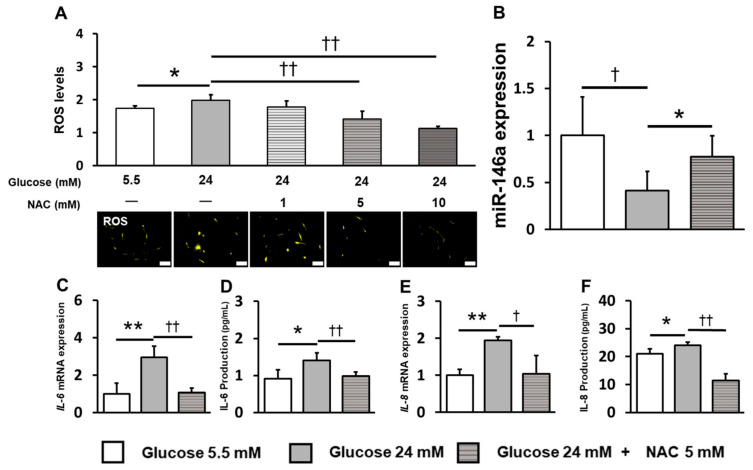
MiR-146a expression and inflammatory cytokine production of hPDLCs are controlled by the production of ROS, which is influenced by NAC. NAC (1, 5 and 10 mM) was added to the medium, and ROS levels were examined under a high glucose concentration. (**A**) Fluorescence staining was performed using a fluorescence microscope. Fluorescence intensity was measured using a microplate reader and the data were compared with those of the control (scale bar: 200 μm). Significant increases were observed compared with those at 5.5 mM glucose: ** p* < 0.05. Significant decreases were observed compared with those at 24 mM glucose: ††*p* < 0.01. (**B**) Expression of miR-146a was measured at 72 h. (**C**) *IL-6* mRNA gene expression was measured at 72 h. (**D**) IL-6 production was measured at 72 h. (**E**) *IL-8* mRNA gene expression was measured at 72 h. (**F**) IL-8 production was measured at 72 h. Significant increases were compared with those at 24 mM glucose: ** p* < 0.05, *** p* < 0.01. Significant decreases were compared with those at 24 mM glucose: †* p* < 0.05, ††* p* < 0.01.

**Figure 5 ijms-25-10702-f005:**
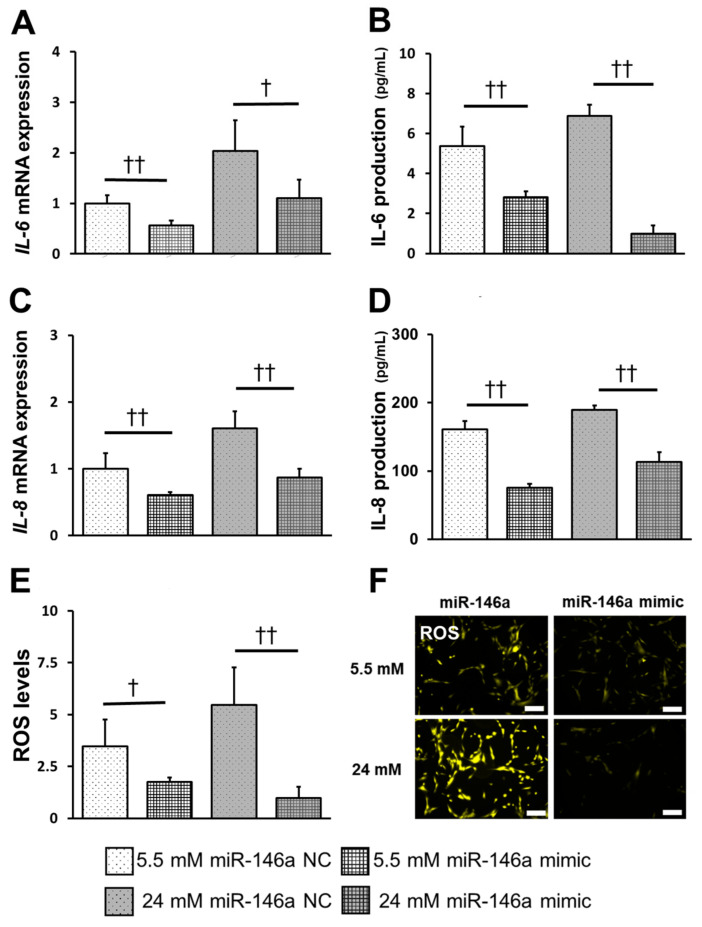
miR-146a down-regulates the inflammatory response and ROS induction in hPDLCs. (**A**,**C**) After 24 h of transfection, cultures were stimulated with 5.5 mM or 24 mM glucose. *IL-6* and *IL-8* mRNA expression was measured at 72 h post-stimulation. (**B**,**D**) After 24 h of transfection, cultures were stimulated with 5.5 mM or 24 mM glucose. IL-6 and IL-8 production was measured at 72 h post-stimulation. (**E**,**F**) After 24 h of transfection, cultures were stimulated with 5.5 mM or 24 mM glucose. Fluorescence staining was performed using a fluorescence microscope. Fluorescence intensity was measured using a microplate reader, and the data were compared with those of the control (scale bar: 200 μm). Significant decreases compared with miR-146a NC: †* p* < 0.05, ††* p* < 0.01.

**Figure 6 ijms-25-10702-f006:**
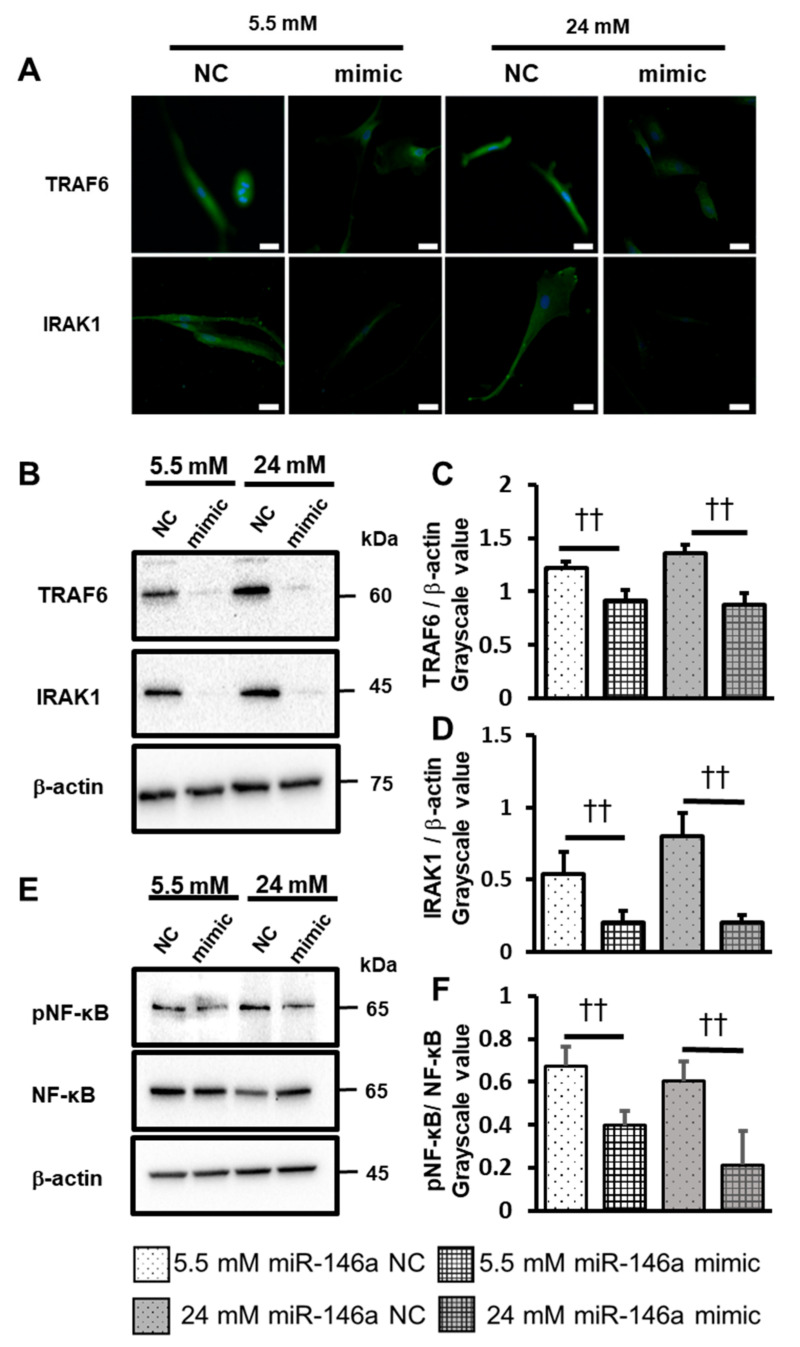
MiR-146a down-regulates IRAK1, TRAF6 and NF-κB in hPDLCs. (**A**) After 24 h of transfection, cultures were stimulated with 5.5 mM or 24 mM glucose. Immunofluorescence staining of IRAK1 and TRAF6 was visualized by confocal laser microscopy after 72 h of incubation (scale bar: 50 μm). (**B**) After 24 h of transfection, cultures were stimulated with 5.5 mM or 24 mM glucose. The levels of IRAK1 and TRAF6 were analyzed using Western blotting. Western blotting was performed on the protein extracts of these cells with antibodies against the indicated proteins, using β-actin as a loading control. (**C**) IRAK1 expression was quantified using ImageJ software. (**D**) TRAF6 expression was quantified using ImageJ software. (**E**) After 24 h of transfection, cultures were stimulated with 5.5 mM or 24 mM glucose. The levels of NF-κB and pNF-κB were analyzed using Western blotting. Western blotting was performed on the protein extracts of these cells with antibodies against the indicated proteins, using β-actin as a loading control. (**F**) NF-κB and p-NF-κB expression was quantified using ImageJ software. Significant decreases compared with miR-146a NC: ††* p* < 0.01.

## Data Availability

The datasets generated and/or analyzed in the current study are available from the corresponding author upon reasonable request.

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
