# Peer review of "MiR-146a Is Mutually Regulated by High Glucose-Induced Oxidative Stress in Human Periodontal Ligament Cells"

_ijms, 2024, doi:10.3390/ijms251910702_

Round 1
Reviewer 1 Report
Comments and Suggestions for Authors This study observed miR-146a expression in hPDLCs under high-concentration glucose conditions, an in vitro model of diabetic periodontitis, and investigated its regulatory signaling pathway and changes in its expression under oxidative stress conditions. Therefore, this study provides important information for understanding miRNA's antioxidant and anti-inflammatory effects on periodontal ligament cells in diabetic periodontitis. However, there are a few suggestions to improve it as follows.;1.Producing reactive oxygen species (ROS), especially superoxide radical (O-), by mitochondria is one of the intracellular mechanisms that reduce the biological availability of NO. The authors need to confirm the RNA expression of iNOS in hPDLCs under high glucose conditions. Also, can you show changes in NO production under high glucose and mirRNA146a mimic conditions?
2. The authors show changes in inflammatory cytokines and ROS under high glucose conditions and changes in inflammatory cytokines and ROS according to the expression level of miRNA146a. These results are not sufficient to support the authors' claims. Therefore, supplementary experiments showing morphological changes in mitochondria and changes in respiration under high glucose conditions seem necessary.
3. In hPDLCs, the mRNA and protein levels of inflammatory cytokines and TNF receptor-associated factor 6 and interleukin-1 receptor-associated kinase 1 were increased under high glucose conditions. However, the expression of NfkB, a regulator of inflammatory factor expression, was not shown. Therefore, the authors should show the expression of NfkB under normal and high glucose conditions in PDLCs. Also, it should be shown whether miRNA146a is involved in NfkB phosphorylation.
4. The expression of inflammatory factors and changes in ROS should be shown in the group in which miRNA146a was suppressed.
Miner comment
1. The concentration calculation of 5.5 mM (99 mg/dL) and 24 mM (432 172 mg/dL) in line 172 is incorrectly indicated. Please recalculate (5.5 mM = 1000 mg/dL; 24 mM = 4320 mg/dL)
2. Add an explanation about diabetic periodontitis in the introduction part.
Reviewer 2 Report
Comments and Suggestions for Authors
Dear authors, thank you for submitting the manuscript "MiR-146a is mutually regulated by high glucose-induced oxidative stress in human periodontal ligament cells". I read your manuscript and here my feedback:
-Please expand the first paragraph. Provide more data how DM is a major factor for periodontal disease. Also describe those "several effects" caused by high glucose conditions.
-Please provide a graphic abstract (GA) displaying the workflow of your study. This can be inserted in the Results section. GA helps the reader to easily understand your study.
-State your hypotheses at the end of the introduction section, and they should be based on previous observations from the literature or your laboratory. In the discussion section, mention if your hypotheses were rejected, accepted or partially rejected/accepted.
-Please do not make short paragraphs out of a single sentence, such as the first one in the Discussion section. You either expand it or join it to the following paragraph.
-At the end of the discussion section please mention the limitations of the study and also what future studies you would like to perform based on your results.
-Informed consent statement is empty.
-Verify the forma of the references, I see different styles.
Comments on the Quality of English Language
Minor revision is needed.
Round 2
Reviewer 1 Report
Comments and Suggestions for Authors
All questions have been resolved.